# Task-based Evaluation of 3D Radial Layouts for Centrality Visualization

### ABSTRACT

In this paper we propose improvements to the 3D radial layouts that make it possible to visualize centrality measures of nodes in a graph. Our improvements mainly relate edge drawing and the evaluation of the 3D radial layouts. First, we projected the edges onto the visualization surfaces in order to reduce the nodes overlap. Secondly, we proposed a human-centered evaluation in order to compare the efficiency score, the time to complete tasks and the number of clicks of the 3D radial layouts to those of the 2D radial layouts. The results showed that even if the overall improvements in terms of time or errors are not statistically significant between the various visualization surfaces, the participants have a better feeling on the 3D and therefore the user experience is able to be improved in data visualization.

**Index Terms:** Human-centered computing—3D graph visualization—Centrality visualization—Layout evaluation

## 1 INTRODUCTION

Centrality measures are topological measures that describe the importance of the nodes in a graph. There has been a lot of work carried out in this topic for network analysis in order to answer the question "Which are the most important nodes in a graph?" [16, 17]. Other works in graph drawing chose to visually reveal these properties in order to facilitate their exploratory analysis [2, 18]. For example, in graph analytics, some works are interested in understanding and describing the interaction structure by analyzing the topology of the graph [6, 21]. Some others are interested in identifying and characterizing the nodes that are particularly important [27] and how their neighbors are connected to each other [29].

However, visualizing these measures in 2D could be difficult when the size of the graph is important in terms of the number of nodes and edges. Indeed, there would be a lot of nodes and edges overlap and edge crossings, which are less of a problem in 3D than 2D [26]. Kobina et al. [13] therefore proposed new 3D methods based on the 2D radial layouts that highlight the centrality of the nodes by optimizing the spatial distribution of the nodes. Nevertheless, in 3D some edges could hide others depending on the position of the observer or the 3D layouts, as can be seen in the proposed methods of Kobina et al. [13] using straight edges.

So, we first propose improvements to the 3D radial layouts by projecting the edges onto the visualization surfaces in order to reduce the nodes overlap. The purpose of our improvements is to provide a better overall view of a complex and large graph than the 3D radial techniques and to reduce the time in exploring and analyzing such a graph. We then propose a task-based evaluation using a well-known centrality measure in order to compare the efficiency score, the time to complete tasks and the number of clicks of the 3D radial layouts to those of the 2D radial layouts. The evaluation tasks are related to the central nodes, to the peripheral nodes and to the dense areas of a graph. The purpose of our evaluation is to show that the 3D radial methods could be better to explore and to analyze graphs whatever the interest, compared to the 2D radial layouts.

This paper is structured as follows: in section 2 we recall some notion about centrality measures in graphs. We review related work on centrality visualization in section 3. Then we present our improvements in section 4 and the human-centered evaluation of these improvements in section 5. In section 6 we present the evaluation results following experiments while in section 7 we present our discussion of the various results. In section 8 we present our conclusion and we finally present our future work in section 9.

## 2 CENTRALITY MEASURES IN GRAPHS

In graph analytics, centrality measures [22] characterize the topological position of the nodes in a graph. In other words, centrality measures make it possible to identify important nodes in the graph and further provide relevant analytical information about the graph and its nodes.

The importance of a node in a graph can be characterized by centrality measures, the clustering coefficient [10] also known as a high density of triangles. Some centrality measures, such as degree centrality, can be computed using local information of the node. The degree centrality quantifies the number of neighbors of a node. Betweenness centrality and closeness centrality [8, 9] use global information of the graph. The betweenness centrality is based on the frequency at which a node is between pairs of other nodes on their shortest paths. In other words, betweenness centrality is a measure of how often a node is a bridge between other nodes. The closeness centrality is the inverse of the sum of distances to all other nodes of the graph.

The clustering coefficient measures to what extent the neighbors of a node are connected to each other. If the neighbors of the node $i$ are all connected to each other, then the node $i$ has a high clustering coefficient.

## 3 CENTRALITY VISUALIZATION

Many works in graph drawing made it possible to convey relational information such as centrality measures and clustering coefficient. So, Brandes et al. [1] and Brandes and Pich [2] proposed radial layouts that make it possible to highlight the betweenness and the closeness centralities of the nodes in a graph. In these methods, each node is constrained to lie on a circle according to its centrality value. Thus, nodes with a high centrality value are close to the center and those of low value are on the periphery.

Dwyer et al. [5] also proposed 3D parallel coordinates, orbit-based and hierarchy-based methods to simultaneously compare five centrality measures (degree, eccentricity, eigenvector, closeness, betweenness). The difference between these three methods is how centrality values are mapped to the node position. So, for 3D parallel coordinates nodes are placed on vertical lines; for orbit-based nodes are placed on concentric circles and for hierarchy-based nodes are placed on horizontal lines. On the other hand, Raj and Whitaker [18] proposed an anisotropic radial layout that makes it possible to highlight the betweenness centrality of the nodes in a graph. In this method, they proposed to use closed curves instead of concentric circles, arguing that the use of closed curves offers more flexibility to preserve the graph structure, compared to previous radial methods.

However, it would be difficult to visually identify some nodes that have the same centrality value, compared to the radial layouts. The proposed methods of Dwyer et al. make it possible to compare many centrality measures, but it would be difficult to identify the central nodes, compared to that of Brandes and Pich. On the other

hand, 2D methods suffer from lack of display space when one needs to display a large graph in terms of number of nodes and edges.

So, Kobina et al. [13] proposed 3D extensions of the radial layouts of Brandes and Pich [2] in order to better handle the visualization of complex and large graphs (see Fig. 1). Their methods consist in projecting 2D graph representations on 3D surfaces. These methods reduce nodes and edges overlap and improve the perception of the nodes connectivity. However, some nodes and edges are less visible depending on the projection surface and edge drawing method. Indeed, the use of straight edges caused some to be inside the half-sphere and others to cross the half-sphere. Furthermore, most of the edges are on the surface for the conical projection and outside the surface for the projection on the torus portion. Some nodes and edges are therefore less visible.

## 4 IMPROVEMENT OF THE 3D RADIAL LAYOUTS

In order to reduce nodes and edges overlap in the proposed methods of Kobina et al. [13], we projected the edges onto the visualization surfaces.

Let $e$ be an edge to be projected onto a visualization surface and that connects nodes $j$ and $k$, and $P_i$ be every point belonging to $e$.

$P_i = P_j + (P_k - P_j)t$ where $P_j$ and $P_k$ are respectively the position of nodes $j$ and $k$, and $t = \frac{i}{n-1}$ where $n$ is the number of control points of the edge $e$.

### 4.1 Edge projection onto the cone

In this section, we describe the various steps that are relevant to the proposed method of projecting edges onto the cone:

- Compute the angle $\theta$ between the x axis and the z axis of the point to be projected: $\theta = \frac{180}{\pi} atan2(z_{P_i}, x_{P_i})$

- Rotate by $\theta$ about y axis. Let $R$ be the rotation result:

$$R = \begin{bmatrix} \cos\theta & 0 & -\sin\theta \\ 0 & 1 & 0 \\ \sin\theta & 0 & \cos\theta \end{bmatrix} \cdot \begin{bmatrix} x \\ y \\ z \end{bmatrix}$$

- Compute the projected point $Proj = \frac{x_{P_i}x_R + y_{P_i}y_R + z_{P_i}z_R}{||R||} \cdot R$

- Compute the altitude $y_{Proj} = 1 - \sqrt{x_{Proj}^2 + z_{Proj}^2}$

### 4.2 Edge projection onto the half-sphere

Here we describe the projection method of the edges onto the half-sphere:

- Compute the projected point $Proj = \frac{P_i}{||P_i||}$

- Compute the altitude $y_{Proj} = \sqrt{1 - (x_{Proj}^2 + z_{Proj}^2)}$

### 4.3 Edge projection onto the torus portion

In this section, we describe the projection method of the edges onto the torus portion in four steps:

- Compute the angle $\theta$ between the x axis and the z axis of $P_i$, the point to be projected: $\theta = \frac{180}{\pi} atan2(z_{P_i}, x_{P_i})$

- Rotate by $\theta$ about y axis. Let $R$ be the rotation result:

$$R = \begin{bmatrix} \cos\theta & 0 & -\sin\theta \\ 0 & 1 & 0 \\ \sin\theta & 0 & \cos\theta \end{bmatrix} \cdot \begin{bmatrix} x \\ y \\ z \end{bmatrix}$$

- Compute the projected point $Proj = \frac{P_i}{||P_i||} + R$

- Compute the altitude of the point:
$y_{Proj} = 1 - \sqrt{1 - ((r-1)(r-1))}$, with $r = \sqrt{x_{Proj}^2 + z_{Proj}^2}$.

Fig. 2 illustrates the result of our projected edges, compared to that of straight edges used in the proposed methods of Kobina et al. [13] (Fig. 1).

Thus, by projecting the edges onto the visualization surfaces, we improved the readability of the graph. Furthermore, there are no edges that cross the visualization surface.

## 5 EVALUATION

We conducted a human-centered evaluation through a series of tasks performed on generated graphs in order to compare the efficiency score, the time to complete a task and the number of clicks of the 3D layouts with projected edges (Fig. 2) to those of the 2D radial layouts. We use these 3 metrics to determine if a kind of visualization is better or worse than the others.

### 5.1 Tasks

Kobina et al. [13] suggested that the projections of the uniform 2D representation highlight either the center, the periphery, or either moderately the center and the periphery. So we chose these three following tasks that are related to the central nodes, to the peripheral nodes and to the dense areas of a graph:

- **Task 1 (related to the central nodes).** The participants were asked to find the node that has the greatest degree among the most central node's neighbors.

- **Task 2 (related to the peripheral nodes).** The participants were asked to find a least central node that has at least 2 neighbors.

- **Task 3 (related to the dense areas of a graph).** The participants were asked to find a node of degree at least 3 and that has the highest clustering coefficient except 100%.

### 5.2 Hypothesis

Based on the proposed methods of Kobina et al. [13], we make the following hypotheses:

**H1.** The 2D that emphasizes the periphery is the worst of the visualization surfaces when one is interested in the central nodes.

**H2.** The 2D that emphasizes the center is the worst of the visualization surfaces when tasks are related to the periphery.

**H3.** The combination of the peripheral emphasis and the different 3D projections highlights not only the peripheral nodes as the 2D peripheral emphasis, but also improves the visibility of the center.

**H4.** The combination of the central emphasis and the different 3D projections highlights not only the central nodes as the 2D central emphasis, but also improves the visibility of the periphery.

**H5.** One spends less time in exploring and analyzing graphs on the 3D surfaces than on the 2D.

**H6.** There are fewer clicks on the 3D surfaces than on the 2D representations.

**H7.** 3D surfaces are better suited for exploring the dense areas of a graph than 2D representations.

### 5.3 Experimental protocol and measures

We conducted an experimental study using a WebGL version of our graph visualization system because of the Covid-19. Here is the link to our experiment for a given configuration: **https://anonymnam.github.io/radialvig3dxp**. Each participant could therefore perform the experiment remotely on his own laptop. Kobina et al. [13] suggested that the combination of the uniform 2D representation and the different projections makes it possible to obtain in addition an emphasis on the center or on the periphery.

spherical projection          conical projection          torus portion

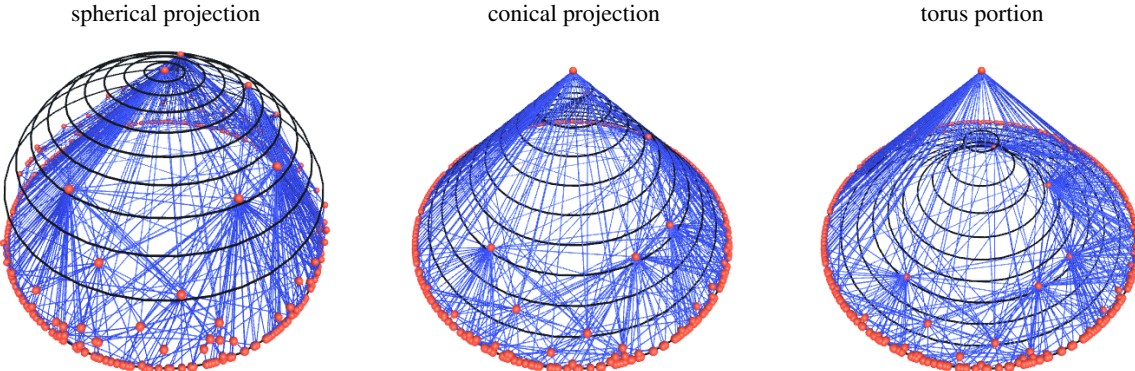

Figure 1: Betweenness centrality: uniform 3D radial visualization (419 nodes and 695 edges). The spherical projection spreads out more the peripheral nodes than the central nodes while the projection on the torus portion spreads out more the central nodes than the peripheral nodes. The conical projection evenly distributes nodes. Images are from [13].

spherical projection          conical projection          torus portion

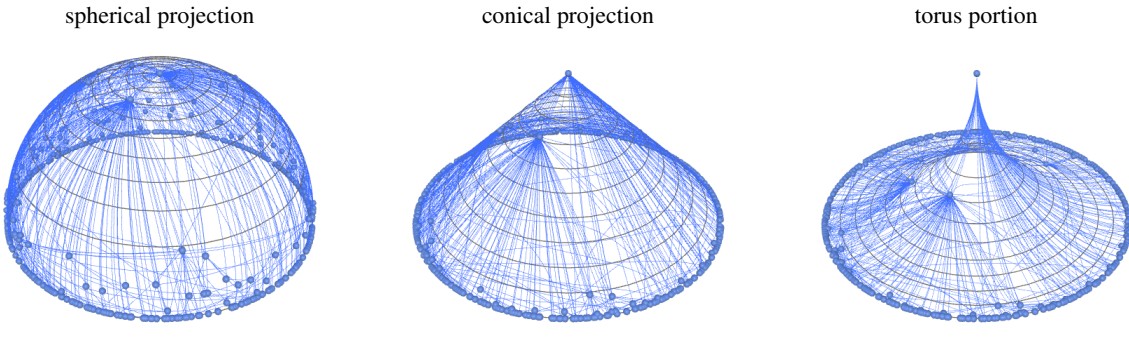

Figure 2: Betweenness centrality: uniform 3D radial visualization (419 nodes and 695 edges). Edges are projected onto the visualization surfaces, compared to straight edges observed in the proposed methods of Kobina et al. [13](Fig. 1).

So in this study, our goal is to show that these 3D methods could be better to explore and to analyze graphs whatever the interest (the central or peripheral nodes, the dense areas), compared to the 2D representations. Indeed, since Kobina et al. [13] optimized the spatial distribution of nodes and we improved the edges drawing by projecting them onto the surfaces, there could be less time in exploration, less clicks and more accurate responses to different tasks, because the perception of the nodes connectivity is improved. Moreover, we want to analyze the usability of the 3D for exploring and analyzing graphs. On the other hand, we want to identify the best layout that could be used to visualize graphs.

For our experiment, we chose to use the betweenness centrality, because it has an interesting use and regardless of the centrality measure, the purpose of the evaluation remains the same. It will therefore be enough to assess the interest of the proposed methods. We first generated, thanks to the Stochastic Block Model algorithm [11, 15, 25], 6 different graphs (250 nodes and 855 edges) that have equivalent topological characteristics (Fig. 3, Fig. 4), since it is difficult to find in databases several graphs of the same size with equivalent topological characteristics (density, clustering coefficient).

The stochastic Block Model is a probabilistic model based on community structure in graphs. This model partitions the nodes in blocks of arbitrary sizes, and places edges between pairs of nodes independently, with a probability that depends on the blocks [24]. Thus, the structure of each community in the graph varies enough to avoid a learning effect.

We then built 24 configurations with the various representation surfaces so that each surface and graph is performed at least once as first, using something similar to the concept of the Latin square [7, 19]. A Latin square is an *n* x *n* array filled with *n* different symbols in such a way that each symbol occurs exactly once in each row and exactly once in each column. For our configurations, we respected a distribution order between 2D and 3D surfaces so that the running order of a 2D representation corresponds to that of the equivalent 3D surface. For example, if a configuration starts with the 2D surfaces and the first surface is the one that emphasizes the center, then the first 3D surface will be the torus portion, since it is the most to highlight the center. So we make sure that each configuration is tested as many times before as after each of the other configurations.

During the experiment and for each task and surface, we measure an efficiency score, the time spent to complete a task and the number of clicks to find an optimal response. As the experiment is done remotely, each participant's performance is automatically saved when he validates his response. Below is how we compute the efficiency score of the participants.

**Task 1.** Find the node that has the greatest degree among the most central node's neighbors.

$$score_i = \begin{cases} 100 * (deg_i/deg_{ideal}), & \text{if } d(ctr, i) = 1 \\ 0, & \text{otherwise} \end{cases} \quad (1)$$

where $deg_i$ is the degree of the selected $node_i$. $deg_{ideal}$ is the greatest degree among the central node's neighbors and $d(ctr, i)$ is

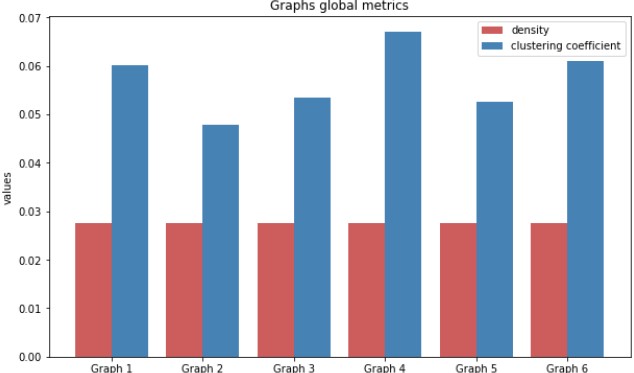

Figure 3: Comparison of generated graphs: all graphs have the same density, but a different clustering coefficient. The clustering coefficient is high if the number of the closed triplets in a graph is important.

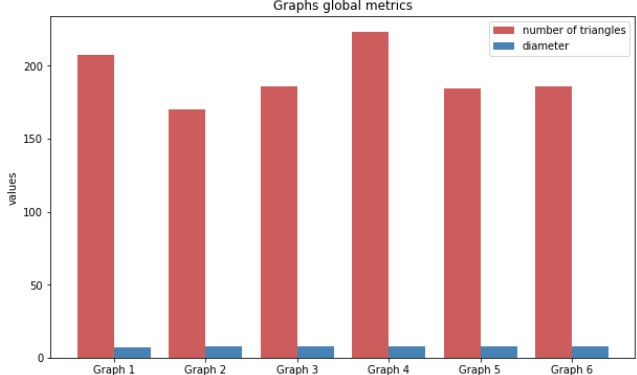

Figure 4: Comparison of generated graphs: all graphs have the same diameter, but different number of triangles. As for the clustering coefficient, the number of triangles is high if the number of the closed triplets in a graph is important.

the shortest distance between the central node and $node_i$. Thus, $node_i$ must be directly connected to the central node, i.e. $d(ctr,i)$ must be equal to 1.

**Task 2.** Find a least central node that has at least 2 neighbors.

$$score_i = \begin{cases} 100 * (1 - c_i)/(1 - c_{ideal}), & \text{if } c_{ideal} \neq 1 \\ 0, & \text{otherwise} \end{cases} \quad (2)$$

where $c_i$ and $c_{ideal}$ are respectively the centrality value of the $node_i$ and that of the ideal node. Furthermore, the score is 0 if the degree of the selected node is less than 2. Indeed, it is easy to check that the degree of the selected node is at least 2. Thus, the score is 0 if the condition is not met. Otherwise, the score varies from 0 at the center to 1 for a node of degree at least 2 and the most on the periphery.

**Task 3.** Find a node of degree at least 3 and that has the highest clustering coefficient except 100%.

$$score_i = \begin{cases} 100 * (ccf_i - ccf_{worst})/d, & \text{if } d > 0 \\ 0, & \text{otherwise} \end{cases} \quad (3)$$

where $d = ccf_{ideal} - ccf_{worst}$. $ccf_i$, $ccf_{worst}$ and $ccf_{ideal}$ are respectively the clustering coefficient of the $node_i$, the worst clustering coefficient and the highest clustering coefficient except 100%. So, the score is 0 if the degree of the selected node is less than 3 or if

the clustering coefficient of the selected node is 100%. Otherwise, we compute the score using equation 3.

At the end of the experiment, each participant completes questionnaires related to the usability of the system and the user experience. Since our experiment is done remotely, we organized a video conference for each participant in order to supervise the experiment's process. The experiment consists of a training phase and an evaluation phase. Before starting the training phase, each participant is instructed about the experiment procedure, its environment, navigation and interaction techniques. For example, when the mouse hovers a node, a tooltip shows its clustering coefficient value and its degree. He is also given the essential notions about graphs in order to ensure that he has the useful knowledge for the experiment. In the training phase, the participant is asked to perform the above tasks on a small graph (the karate club's graph [28]) and on each surface. Once familiar with the system, he moves on to the evaluation phase, but with generated graphs. If the participant is ready to start the training or the evaluation, he clicks on a start button to see the first task to complete and the next task is automatically displayed after validating the previous task's response.

### 5.4 Participants

For this project, we were needing a number of participants that would be a multiple of 24 in order to encounter the same number of these 24 configurations mentioned above. So, there were 24 participants (9 female, 15 male) and they were recruited among our colleagues in the laboratory and among students: 50% are between 18 and 25 years old, 37.5% are between 25 and 35, and 12.5% are more than 35 years old. Moreover, most participants had no experience in data analysis and data visualization, but some of them had gaming experience.

## 6 RESULTS

### 6.1 User performance

We present here the main results from the analysis of the data collected during our experiment through nonparametric tests using the Kruskal method [14] and post-hoc tests using the Dunn's method [4, 20]. We used nonparametric tests since none of the samples comes from a normal distribution (normality tests were done using the Shapiro-Wilk method [23]). As a reminder, the variables analyzed are the efficiency score, the time and the number of clicks for each task and each surface.

#### 6.1.1 Task 1: Find the Node that has the Greatest Degree among the most Central Node's Neighbors

**Efficiency score.** After an exploratory data analysis using box plots (Fig. 5), the nonparametric test showed that there is a statistically significant difference between the visualization surfaces and cannot be due to chance ($F - statistic = 31.46, p = 0.000 < 0.05$). So, we rejected the null hypothesis that the efficiency score is the same for all the visualization surfaces when one is interested in the central nodes. The result of the multiple pairwise comparison (Table 1) showed that the 2D that emphasizes the periphery had a difference of medians.

From the statistic test results, **we validate hypothesis H1** that the 2D representation that emphasizes the periphery is worse to perform tasks that are related to the central nodes, compared to all other surfaces. However, **we validate hypothesis H3** that the 3D projections of the peripheral emphasis not only give the same benefit on the periphery, but also provide the visibility of the center.

**Time.** Considering the results of Fig. 6 we could say that the participants spent more time on the 2D that emphasizes the periphery, compared to all other visualization surfaces. However, we failed to reject the hypothesis of equality of medians ($F - statistic = 5.990, p = 0.307 > 0.05$). From our exploratory analysis results,

Table 1: Task 1: Efficiency score: P-values of the multiple pairwise comparison using Dunn's method (significant p-values starred (* $p < 0.05$, ** $p < 0.01$, *** $p \leq 0.001$)).

|  | 2D central | 2D peripheral | 2D uniform | Cone | Half sphere | Torus |
|---|---|---|---|---|---|---|
| 2D central | 1 | 0.00002*** | 1 | 1 | 1 | 1 |
| 2D peripheral | 0.00002*** | 1 | 0.0008*** | 0.00013*** | 0.00067*** | 0.00072*** |
| 2D uniform | 1 | 0.0008*** | 1 | 1 | 1 | 1 |
| Cone | 1 | 0.00013*** | 1 | 1 | 1 | 1 |
| Half sphere | 1 | 0.00067*** | 1 | 1 | 1 | 1 |
| Torus | 1 | 0.0007*** | 1 | 1 | 1 | 1 |

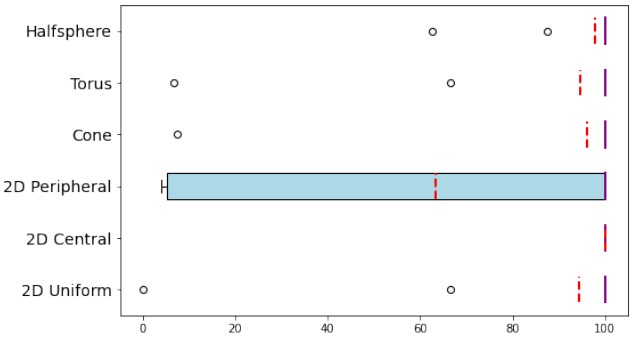

Figure 5: Task 1: Efficiency score: Descriptive representation (mean in red dashes, median in purple). The low mean of the 2D that emphasizes the periphery shows that the participants had a low efficiency score on this surface, compared to all other surfaces.

we cannot validate hypothesis H1, so we cannot prove that the 2D that emphasizes the periphery is worse to perform a task that is related to the central node, compared to all other visualization surfaces. Moreover, we reject hypothesis H5 that the participants spend less time on the 3D surfaces, compared to the 2D surfaces.

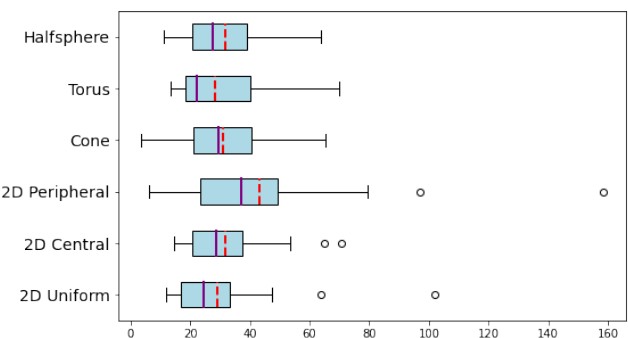

Figure 6: Task 1: Time: Descriptive representation (mean in red dashes, median in purple). Here, we could say that the participants spent more time on the 2D that emphasizes the periphery, compared to all other visualization surfaces.

Number of clicks. From the results of Fig. 7, we validate hypothesis H6 that the participants clicked less on the 3D surfaces, compared to the 2D representations. Furthermore, the nonparametric test result showed that there is a statistically significant difference between the visualization surfaces, since the F-statistic is 12.554 and the corresponding p-value is $0.028 < 0.05$. So we conclude that the type of surface leads to statistically significant differences in the number of clicks. A multiple pairwise comparison (Table 2) confirmed our exploratory analysis that the 2D that emphasizes the periphery is

different from the other surfaces.

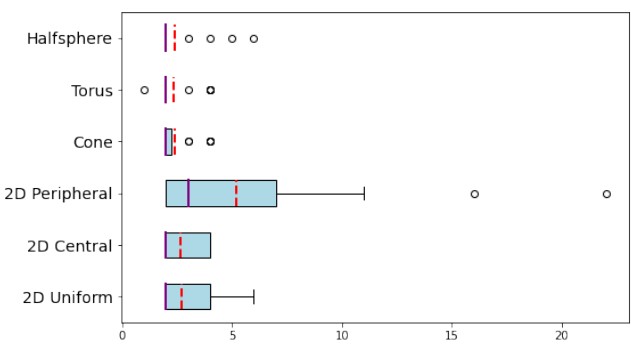

Figure 7: Task 1: Number of clicks: Descriptive representation (mean in red dashes, median in purple). We could suppose that the participants clicked less on the 3D surfaces, compared to the 2D.

Unlike the score analysis, there is a statistically significant difference between the 2D that emphasizes the periphery and two 3D surfaces (the half-sphere and the torus portion).

Ultimately, the 3D surfaces are well suited for carrying out tasks that are related to the central nodes because Fig. 7 and Table 2 show that our hypothesis H6 is validated for the number of clicks. Moreover, we validated hypothesis H1 and H3 for the efficiency score. However we cannot prove that our hypotheses H1 and H5 could be validated with respect to the time of the task.

### 6.1.2 Task 2: Find a least Central Node that has at least 2 Neighbors

Efficiency score. From an exploratory analysis (Fig. 8) we validate hypothesis H2 that the 2D representation that emphasizes the center is worse when a task is related to the peripheral nodes, compared to all other visualization surfaces, since the participants did not have good scores on the 2D that emphasizes the center. Moreover, there is a difference that is statistically significant between the 2D that emphasizes the center and all the other surfaces (see Table 3), because the F-statistic is 40.31 and the corresponding p-value is $0.000 < 0.05$. Nonetheless, we validate hypothesis H4 that the 3D projections of the central emphasis make it possible not only to get the same visual effect on the center, but also to improve the visibility of the periphery.

Time. As far as the time analysis is concerned, we could say that the participants spent less time on the 3D surfaces and the uniform 2D, compared to the 2D surfaces that emphasize the center and the periphery (Fig. 9). However, we failed to reject the hypothesis of equality of medians ($F - statistic = 1.65, p = 0.90 > 0.05$). So, we reject hypothesis H5 that the participants spent less time on the 3D surfaces.

Number of clicks. Fig. 10 shows high values of medians and means for the 2D that emphasizes the periphery and the cone, compared to

Table 2: Task 1: Number of clicks: P-values of the multiple pairwise comparison using Dunn's method (significant p-values starred (* p < 0.05, ** p < 0.01, *** p ≤ 0.001)).

|  | 2D central | 2D peripheral | 2D uniform | Cone | Half sphere | Torus |
|---|---|---|---|---|---|---|
| 2D central | 1 | 0.679 | 1 | 1 | 1 | 1 |
| 2D peripheral | 0.679 | 1 | 0.517 | 0.13 | 0.0435* | 0.0377* |
| 2D uniform | 1 | 0.517 | 1 | 1 | 1 | 1 |
| Cone | 1 | 0.13 | 1 | 1 | 1 | 1 |
| Half sphere | 1 | 0.0435* | 1 | 1 | 1 | 1 |
| Torus | 1 | 0.0377* | 1 | 1 | 1 | 1 |

Table 3: Task 2: Efficiency score: P-values of the multiple pairwise comparison using Dunn's method (significant p-values starred (* p < 0.05, ** p < 0.01, *** p ≤ 0.001)).

|  | 2D central | 2D peripheral | 2D uniform | Cone | Half sphere | Torus |
|---|---|---|---|---|---|---|
| 2D central | 1 | 0.00000*** | 0.002** | 0.00003*** | 0.00000*** | 0.001*** |
| 2D peripheral | 0.00000*** | 1 | 1 | 1 | 1 | 1 |
| 2D uniform | 0.002** | 1 | 1 | 1 | 0.752 | 1 |
| Cone | 0.00003*** | 1 | 1 | 1 | 1 | 1 |
| Half sphere | 0.00000*** | 1 | 0.752 | 1 | 1 | 0.986 |
| Torus | 0.001*** | 1 | 1 | 1 | 0.986 | 1 |

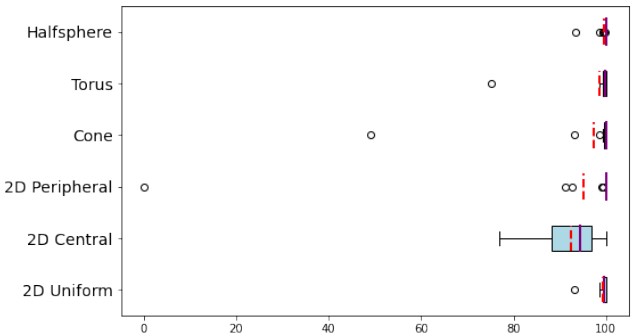

Figure 8: Task 2: Efficiency score: Descriptive representation (mean in red dashes, median in purple). We could say that the participants did not have good scores on the 2D that emphasizes the center, compared to all other surfaces.

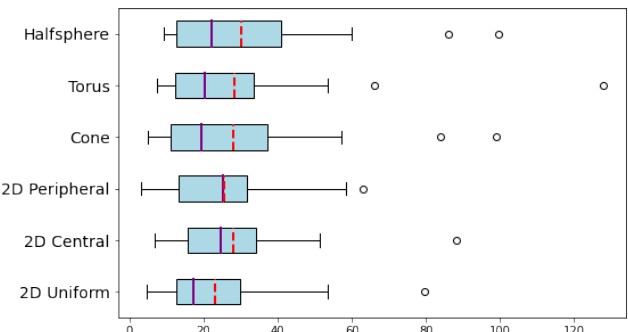

Figure 9: Task 2: Time: Descriptive representation (mean in red dashes, median in purple). We could say that the participants spent less time on the 3D surfaces and the uniform 2D.

all other surfaces. So, it could suggest that the participants clicked more on the 2D that emphasizes the periphery and on the cone. On the other hand, the nonparametric test failed to reject the hypothesis of median equality ($F-statistic = 2.93, p = 0.71 > 0.05$). So, as

for the time analysis, the difference in medians observed could suggest that the 2D that emphasizes the periphery and the cone are worse when one is interested in the peripheral nodes. So, **we reject hypothesis H6** that there are fewer clicks on the 3D surfaces.

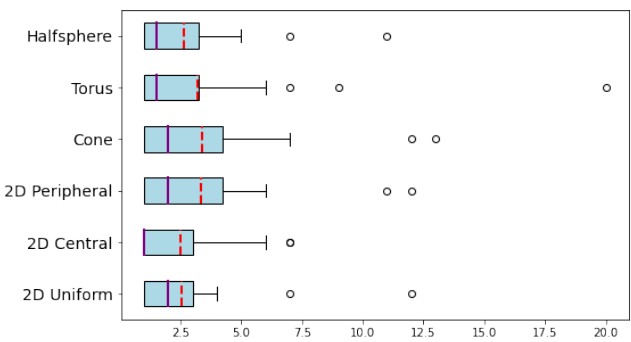

Figure 10: Task 2: Number of clicks: Descriptive representation (mean in red dashes, median in purple). It could suggest that the participants clicked more on the 2D that emphasizes the periphery and on the cone.

Based on the various analyses of task 2, that of the efficiency score makes it possible to validate hypotheses **H2** that the 2D that emphasizes the center is the worst of the visualization surfaces when tasks are related to the peripheral nodes, and **H4** that our 3D projections make it possible not only to get the same benefit on the center, but also to improve the visibility of the periphery. Furthermore, Table 3 shows that the half-sphere and the cone are well suited when one is interested in the peripheral nodes. However, analyses of time and number of clicks show that the difference in medians could suggest that the 2D that emphasizes the periphery is worse, compared to other surfaces and that the 3D surfaces are better, but we cannot prove that hypotheses **H5** and **H6** could be validated.

### 6.1.3 Task 3: Find a Node of Degree at least 3 and that has the Highest Clustering Coefficient except 100%

**Efficiency score.** From an exploratory analysis results (Fig. 11) we could suppose that the participants got good scores on the 2D that

emphasizes the center. On the other hand, we failed to reject the null hypothesis that the efficiency score is the same for all the visualization surfaces, since the test statistic is 6.0 and the corresponding p-value is $0.31 > 0.05$. So, the difference in medians could lead us to say that the 2D that emphasizes the center is better for exploring the dense areas of the graph, compared to all other surfaces and that our hypothesis **H7** (3D surfaces are better suited for exploring the dense areas of a graph) is rejected, but the statistic analysis failed to demonstrate it.

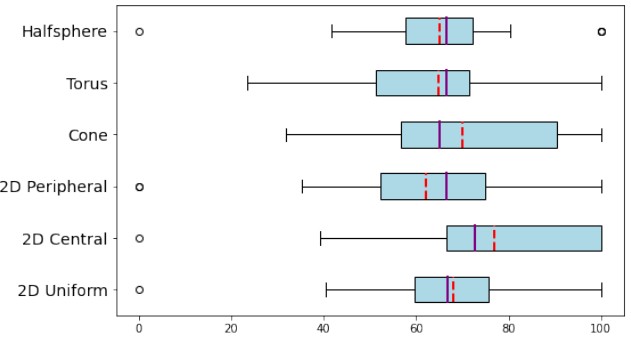

Figure 11: Task 3: Efficiency score: Descriptive representation (mean in red dashes, median in purple). We could suppose that the participants got good efficiency scores on the 2D that emphasizes the center.

**Time.** As for the score analysis, Fig. 12 shows the result of an exploratory data analysis that could lead one to think that the participants spent less time on the uniform 2D, compared to all other surfaces. However, the median values are not significantly different, since the nonparametric test did not reject the hypothesis of median equality ($F - statistic = 1.04, p = 0.96 > 0.05$). So, **we reject hypotheses H5** that the participants spend less time on the 3D surfaces and **H7** that the 3D surfaces are better than the 2D surfaces to explore the dense areas of a graph.

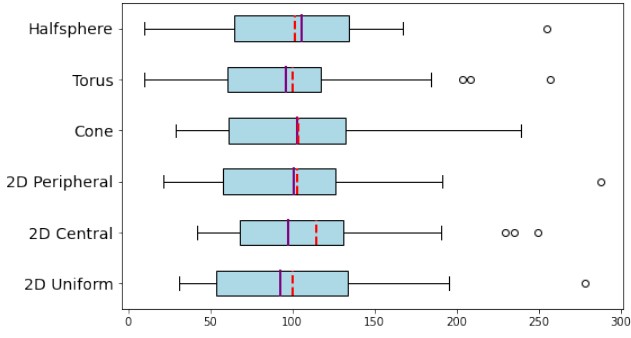

Figure 12: Task 3: Time: Descriptive representation (mean in red dashes, median in purple). We could suppose that participant spent less time on the uniform 2D, compared to other surfaces.

**Number of clicks.** Fig. 13 shows that the median value of the torus portion is smaller than the median values of other visualization surfaces. So we could say that the participants clicked less on the torus portion, compared to all other surfaces. **We could then validate hypothesis H6** that there are less clicks on the 3D surfaces. However, we failed to reject the null hypothesis that the number of clicks is the same for all the visualization surfaces when tasks are related to the dense areas ($F - statistic = 5.0, p = 0.42 > 0.05$). So, **we reject hypothesis H6** that there are fewer clicks on the 3D

surfaces, and **hypothesis H7** that the 3D surfaces are better suited than the 2D to explore the dense areas of a graph.

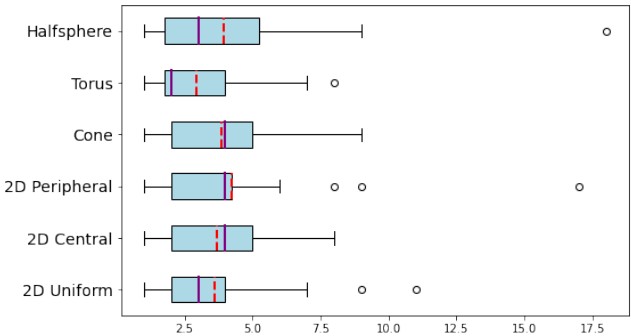

Figure 13: Task 3: Number of clicks: Descriptive representation (mean in red dashes, median in purple). It could suggest that the participants clicked less on the torus portion, compared to all other surfaces.

Unlike the various analyses carried out for tasks 1 and 2, those of task 3 showed in exploratory analysis that some 3D visualization surfaces are better than the 2D surfaces, but the statistic tests showed that the differences of medians in efficiency score, in time and in number of clicks are not statistically significant when one is interested in the dense areas of the graph. So, **we reject hypotheses H5** that the participants spend less time in exploring and analyzing graphs on the 3D surfaces, **H6** that there are fewer clicks on the 3D surfaces and **H7** that the 3D surfaces are better than the 2D to explore the dense areas of a graph.

### 6.2 User experience

As mentioned above (in Sect. 5.3), at the end of the experiment, the participants were asked to complete a questionnaire related to the system usability and to their experience. As far as their experience is concerned, they were asked whether they understood the requested tasks, if they had difficulty interacting with the system, and if they had visual fatigue. The results were that 23 participants over 24 understood the requested tasks, 7 over 24 had difficulty interacting with the system and 7 participants over 24 declared having visual fatigue.

The participants were also asked to specify the surfaces that enabled them to better perform the requested tasks, on the one hand, and to identify the surfaces with which they had difficulty completing the requested tasks, on the other hand. Based on their feedback, 3D surfaces have significantly contributed to the successful completion of the various tasks, compared to the 2D representations (uniform 2D, the 2D that emphasizes the center or the periphery). Fig. 14 and Fig. 15 illustrate the distribution of user preferences for a successful and unsuccessful completion, respectively. Moreover, Fig. 14 shows that the 2D that emphasizes the center and the 2D that emphasizes the periphery alone total 80% of votes while the cone makes 0%.

### 7 DISCUSSION

Some nodes would be less visible with the use of the straight edges in the proposed methods of Kobina et al. [13]. Indeed, combining the peripheral emphasis and the projection of the nodes and edges on the half-sphere or on the torus portion, some intermediate nodes would be less visible due to the surface, unlike the conical projection. Furthermore, with uniform projections, some nodes and edges would be less visible in the dense areas according to the projection surface. So, projecting the edges onto the visualization surface, we reduced the overlap of the nodes and the edges, and we therefore improved the overall readability of the graph.

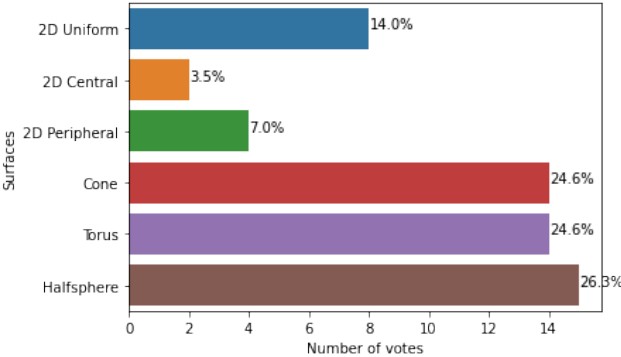

Figure 14: Surfaces that the participants prefer when performing tasks.

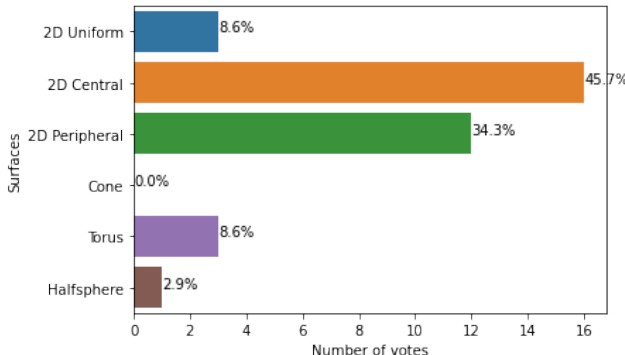

Figure 15: Surfaces that the participants do not like when performing tasks.

As far as our evaluation is concerned, the results did not allow us to identify which representation is best suited to visualize large graphs and to improve graph analysis. However, **we partially validated hypotheses H1, H2, H3, H4 and H6**, since some statistic test results showed that there are differences in efficiency score and in number of clicks.

Indeed, these results made it possible to **validate hypotheses H1** that the 2D that emphasizes the periphery is the worst of the surfaces to visualize the center, and **H2** that the 2D that emphasizes the center is the worst of the surfaces to visualize the periphery with respect to the efficiency score of tasks 1 and 2. Moreover, **we validated hypotheses**: 1) **H3** that the combination of the peripheral emphasis and different 3D projections makes it possible not only to get the same advantages on the periphery as the 2D peripheral emphasis, but also to improve the visibility of the center; 2) **H4** that combining the central emphasis and different 3D projections makes it possible not only to get the same benefits on the center as the 2D central emphasis, but also to improve the visibility of the periphery, always regarding the efficiency score of tasks 1 and 2. **We also validated hypothesis H6** that there are fewer clicks on the 3D surfaces regarding the number of clicks of task 1.

On the other hand, **we rejected hypotheses H5** and **H7**, since we were not able to prove that: 1) participants spend less time on the 3D surfaces and 2) the 3D surfaces are better than the 2D to explore the dense areas of a graph. We could therefore say that the 2D versus 3D debate still persists [3]. On the other hand, participants' feedback showed that the 3D surfaces could be well suited for completing the various requested tasks successfully, compared to the 2D surfaces.

## 8 CONCLUSION

In this work, we improve the edge drawing of some 3D graph visualization methods previously proposed. Our improvements consist in projecting the edges onto each visualization surface in order to reduce the nodes and edges overlap.

An online human-centered experimental study was conducted in order to compare the efficiency score, the time to complete tasks and the number of clicks of the various visualization surfaces. We showed through our experiment that there is no difference that is statistically significant in terms of time or errors between these surfaces. However, the participants have a better feeling on the 3D when carrying out the requested tasks, compared to the 2D layouts. Thus, adding a third dimension to the 2D radial views improves the user experience.

## 9 FUTURE WORK

In the future, we will also study in detail the results obtained with large graphs in order to check whether current trends are confirmed. Moreover, we projected the 2D views on other types of 3D surfaces (a parabola, a Gaussian, a hyperboloid and a square root). Thus, we will study in more details the results of these contributions in order to identify the most appropriate approach or combination of approaches that could be used to visualize large and complex graphs.

In order to declutter graphs in the proposed methods of Kobina et al. [13], we have already implemented the kernel density estimation edge bundling algorithm using computer graphics acceleration techniques. Fig. 16 illustrates the result of a graph which was generated using Stochastic Block Model algorithm presented in section 5.3. So, with the bundled graph, it is possible to see how groups of nodes are connected to each other, compared to the unbundled graph. However, we lose the detailed connectivity of a node (for instance, edges between a node and its neighbors). It could be therefore useful to combine the bundled and the unbundled edges for further analysis if one would need to switch between detailed and bundled views.

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

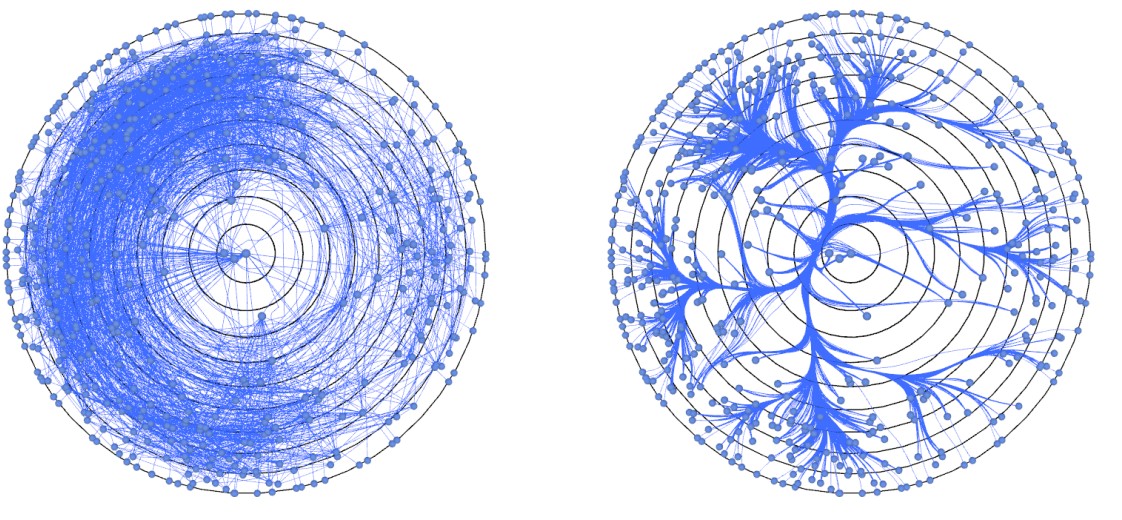

|  unbundled  |  bundled using KDEEB algorithm proposed by Hurter et al. [12]  |

Figure 16: Top view from the cone of a generated graph (500 nodes and 3294 edges). Edge bundling makes it possible to declutter the graph.

collections of connections. In D. L. Hansen, B. Shneiderman, M. A. Smith, and I. Himelboim, eds., *Analyzing Social Media Networks with NodeXL (Second Edition)*, pp. 31–51. Morgan Kaufmann, second edition ed., 2020. doi: 10.1016/B978-0-12-817756-3.00003-0

[11] P. W. Holland, K. B. Laskey, and S. Leinhardt. Stochastic blockmodels: First steps. *Social Networks*, 5(2):109–137, 1983. doi: 10.1016/0378-8733(83)90021-7

[12] C. Hurter, O. Ersoy, and A. Telea. Graph bundling by kernel density estimation. *Computer Graphics Forum*, 31:865–874, 06 2012. doi: 10.1111/j.1467-8659.2012.03079.x

[13] P. Kobina, T. Duval, and L. Brisson. 3d radial layout for centrality visualization in graphs. In L. T. D. Paolis and P. Bourdot, eds., *Augmented Reality, Virtual Reality, and Computer Graphics - 7th International Conference, AVR 2020, Lecce, Italy, September 7-10, 2020, Proceedings, Part I*, vol. 12242 of *Lecture Notes in Computer Science*, pp. 452–460. Springer, 2020. doi: 10.1007/978-3-030-58465-8_33

[14] W. H. Kruskal and W. A. Wallis. Use of ranks in one-criterion variance analysis. *Journal of the American Statistical Association*, 47(260):583–621, 1952.

[15] C. Lee and D. J. Wilkinson. A review of stochastic block models and extensions for graph clustering. *Applied Network Science*, 4(1), Dec 2019. doi: 10.1007/s41109-019-0232-2

[16] F. Martino and A. Spoto. Social network analysis: A brief theoretical review and further perspectives in the study of information technology. *PsychNology Journal*, 4:53–86, 01 2006.

[17] R. Y. Nooraie, J. E. M. Sale, A. Marin, and L. E. Ross. Social network analysis: An example of fusion between quantitative and qualitative methods. *Journal of Mixed Methods Research*, 14(1):110–124, 2020. doi: 10.1177/1558689818804060

[18] M. Raj and R. T. Whitaker. Anisotropic radial layout for visualizing centrality and structure in graphs. In F. Frati and K. Ma, eds., *Graph Drawing and Network Visualization - 25th International Symposium, GD 2017, Boston, MA, USA, September 25-27, 2017, Revised Selected Papers*, vol. 10692 of *Lecture Notes in Computer Science*, pp. 351–364. Springer, 2017. doi: 10.1007/978-3-319-73915-1_28

[19] J. T. Richardson. The use of latin-square designs in educational and psychological research. *Educational Research Review*, 24:84–97, 2018. doi: 10.1016/j.edurev.2018.03.003

[20] L. Sangseok and L. D. Kyu. What is the proper way to apply the multiple comparison test? *Korean J Anesthesiol*, 71(5):353–360, 2018. doi: 10.4097/kja.d.18.00242

[21] M. Saqr, U. Fors, and J. Nouri. Using social network analysis to understand online problem-based learning and predict performance. *PLOS ONE*, 13(9):1–20, 09 2018. doi: 10.1371/journal.pone.0203590

[22] A. Saxena and S. Iyengar. Centrality measures in complex networks: A survey. *ArXiv*, abs/2011.07190, 2020.

[23] S. S. Shapiro and M. B. Wilk. An analysis of variance test for normality (complete samples). *Biometrika*, 52(3/4):591–611, 1965.

[24] T. A. Snijders and K. Nowicki. Estimation and prediction for stochastic blockmodels for graphs with latent block structure. *Journal of Classification*, 14(1):75–100, Jan 1997. doi: 10.1007/s003579900004

[25] N. Stanley, T. Bonacci, R. Kwitt, M. Niethammer, and P. Mucha. Stochastic block models with multiple continuous attributes. *Applied Network Science*, 4:1–22, 08 2019. doi: 10.1007/s41109-019-0170-z

[26] A. R. Teyseyre and M. R. Campo. An overview of 3d software visualization. *IEEE Transactions on Visualization and Computer Graphics*, 15(1):87–105, 2009. doi: 10.1109/TVCG.2008.86

[27] J. Wang, X. Hou, K. Li, and Y. Ding. A novel weight neighborhood centrality algorithm for identifying influential spreaders in complex networks. *Physica A: Statistical Mechanics and its Applications*, 475:88–105, 2017. doi: 10.1016/j.physa.2017.02.007

[28] W. W. Zachary. An information flow model for conflict and fission in small groups. *Journal of anthropological research*, 33:452–473, 1977.

[29] H. Zhang, Y. Zhu, L. Qin, H. Cheng, and J. X. Yu. Efficient local clustering coefficient estimation in massive graphs. In S. Candan, L. Chen, T. B. Pedersen, L. Chang, and W. Hua, eds., *Database Systems for Advanced Applications*, pp. 371–386. Springer International Publishing, Cham, 2017.

