# OpenReview forum: "$Task-based Evaluation of 3D Radial Layouts for Centrality Visualization$"
_graphicsinterface.org/Graphics_Interface/2022/Conference — Submitted to GI 2022_

### Official Review · Reviewer_XCSW · 2022-01-12
**Needs improvements**

**Rating:** 4
**Confidence:** 4

**Review:**

The paper discusses methods used to increase the readability of graphs with a high number of nodes and edges by projecting edges onto the surfaces of graph visualizations. Later, they evaluated the new design and compared the usability of the new layout with the 2D radial layouts.
Three tasks were defined to assess the proposed technique of visualizing nodes and links on a graph. Additionally, seven hypotheses were tested in the study. However, the hypotheses were framed vaguely and needed refinement. For example, H1 mentions that 2D that emphasizes the periphery is the worst of the visualization surfaces; however, it is not clear worse in what sense? Performance? Crowdedness? Edge or node overlap? The results of the study with 24 participants showed some improvements in comparison to the 2D visualization methods.
The paper has small contrition. The writing needs a lot of improvements. The work is not placed well in comparison to the existing body of literature in this area; rather, it is mainly compared with only one other technique. Overall, the paper is not ready for publication at this stage.

Minor edits:
Pg1, last sentence: Brandes and Pich’s citation is missing.
Section 5.3 “He is also given the essential notions about graphs in order to ensure that he has the useful knowledge for the experiment.” Please use neutral pronouns, rather than “He”; you had 9 female participants, why are you only referring to participants with a “he”?

---

### Official Review · Reviewer_XzpW · 2022-01-15
**Interesting paper but the contributions are somewhat incremental and and further qualitative insights are desired**

**Rating:** 5
**Confidence:** 3

**Review:**

This paper focuses on improving 3D radial layouts with a goal to visualize centrality measures of nodes in a graph. More specifically, the authors reduced the overlap of nodes by  improving edge drawing of a 3D layout generation method from Kobina et al. The authors then evaluated their 3D radial layouts by comparing them with the  2D radial layout counterparts.

Strengths:
- Interesting improvement to 3d layouts for graph visualization.
- The study is reasonably well designed and some experimental results suggest the usefulness of the method.

Weaknesses:
-  Novelty of the graph drawing technique seems less convincing as it is based on a prior method Kobina et al.
- The introduction seems somewhat weak without the visual example of a graph that would motivate the problem. Similarly, it remains somewhat unclear what are the key contributions of the work.
- Additional information on the study including the examples of 2D visualizations that were compared with would be helpful for improving the reproducibility of the work.
- In general, 3D visualizations are known to suffer from perceptual issues like occlusion problems etc. Further discussions on the challenges  including more qualitative insights from the study would be helpful

---

### Official Review · Reviewer_u9Bu · 2022-01-16
**overall an interesting study, yet it's realisation leaves too many open questions and the impact remains unclear**

**Rating:** 4
**Confidence:** 4

**Review:**

Summary: This work evaluates 3D radial layouts for visualizing centrality measures in graphs. It mainly builds on previous work from Kobina et al. who introduced new methods to project 2D graph representations on three 3D surfaces: a half-sphere, a cone, and a torus portion. While Kobina et al. used straight edges between the nodes, the author(s) proposed projecting the edges onto the visualization surfaces to reduce node and edges overlap. An experimental study with 24 participants was conducted on a testbed to evaluate 2D and 3D radial graph layouts based on three tasks focusing on central, peripheral, or both nodes respectively.

The paper proposes an interesting approach for visualizing centrality measures of nodes in graphs. The projection of edges onto the visualization surfaces seems to improve the interpretability of large graphs. It is highly appreciated that the author(s) provided a WebGL version of the testbed, which made it easy to clearly understand the experiment procedure as well as the tasks and it allowed to see the generated graph used for the experiment. However, it would be nice if the six different views of a graph that are being compared are shown at least once in the paper.

However, there are several issues with this work:
-	The overall contribution of the paper is not very strong. A new approach of projecting the edges onto the visualization surfaces has been proposed but the evaluation lacks insight.
-	The connection between the tasks and hypothesis should be stated more clearly: Which hypotheses are relevant for which tasks and variables (efficiency score, time, and number of clicks). A summary table could be helpful.
-	The generated graph used in the testbed has 250 nodes and 855 edges – how did the author(s) arrive at this graph size? This should be well justified since different sizes of graphs could influence what graphical representations work best.
-	The appearance of Fig 1+2 is a bit misleading when they are compared. Fig 1 uses red nodes and more salient lines. It seems that Fig 2 uses lines with a certain opacity.
-	The paper is mostly easy to read but it is not always easy to follow. The result section was especially hard to follow since evaluating seven hypotheses relating to three tasks each having three measures needed a lot going back and forth in the paper. A summary table to see which hypotheses have been validated/rejected for which task/measure would have been helpful.
-	One of the measures captured is the number of clicks of a study participant during the tasks. The definition of a click is still unclear: Is rotating the view with the right mouse button considered as a click or just when the user selects a node in the graph? This is an important information when comparing 2D and 3D representations since rotating might not be necessary on the 2D representations, but it is important for the 3D surfaces (since each representation starts with a 2D view by default).
-	The experimental study was done remotely due to COVID-19. Each participant performed the task on their own device. It would be important to know what screen sizes the participants' monitors had since this could influence the result of the user experience.
-	Each of the 24 participants seems to have completed the tasks with the 2D/3D representations in the same order. It would have been better to present the 2D representations followed by the 3D representations to half of the participants and the other way around for the second half of participants to avoid priming.
-	The structure of the paper should be optimized: The layout of the figures in the result section is random which makes it hard to relate them to the text.
-	The visualization used in the result figure is unclear. Some use box-plots and some don't: why is this? (see, e.g., Fig 5, 2D Peripheral uses box plots and the others don't)
-	The authors state "As for the score analysis, Fig. 12 shows the result of an exploratory data analysis that could lead one to think that the participants spent less time on the uniform 2D, compared to all other surfaces." However, looking at Fig 12 the timings between the different interfaces seem comparable, i.e. without a preference for any particular one.

Further suggestions:
-	Instead of using yes/no-questions in the questionnaire (e.g., “the participants were asked if they had difficulty interacting with the system”), the author(s) could use the system usability scale (SUS) by John Brooke to evaluate the usability of the tool.
-	Evaluating large and complex tasks by using edge bundling on the surface seems promising – the user could be given the opportunity to select a threshold for edge bundling to adjust the strength of the edge bundles in real-time during the analysis.
-	The writing needs some improvement.

Minor mistake in writing:
-	“For example, if a configuration starts with the 2D surfaces and the first surface is the one that emphasizes the center, then the first 3D surface will be the torus portion since it is the most [word is missing] to highlight the center.”

Some issues with references:
-	please use proper capitalization: [6]

---

### Decision · Program_Chairs · 2022-01-18

Reject